# Non-Replacement Function Space Sampling for Bayesian Optimization

## Abstract

Bayesian optimization (BO) is a probabilistic framework for global optimization of expensive black-box functions, typically guided by an acquisition function that balances exploration and exploitation. We propose a novel acquisition strategy—Non-Replacement Function Space Sampling (NRFS). Instead of explicitly balancing the exploration–exploitation trade-off as in traditional BO methods, NRFS implicitly achieves this balance by prioritizing sampling functions from the function space that have not been involved in previous acquisition decisions. By establishing a correspondence between each candidate and the set of functions that consider it as the corresponding optimizer, we derive a principled and efficient searching strategy in the design space. We provide strong empirical evidence demonstrating that NRFS achieves state-of-the-art performance across a range of benchmark tasks, consistently improving optimization performance in all settings, particularly in challenging settings that demand both broad exploration and precise exploitation.

## 1 Introduction

Bayesian Optimization (BO) is one of sample-efficient strategies to optimize black-box functions that are often expensive to query. BO strives for the balance between exploration and exploitation to efficiently identify optimal solutions (Jalali et al., 2012; Candelieri, 2023), where exploration encourages querying in regions of high uncertainty to improve understanding of function responses globally, while exploitation favors regions with promising predicted values to quickly find optima (de Ath et al., 2021). Classical acquisition functions in BO, such as Expected Improvement (EI) (Jones et al., 1998) and Probability of Improvement (PI) (Kushner, 1964; Snoek et al., 2012), primarily focus on quantifying potential performance gains. Entropy-based approaches, including Predictive Entropy Search (PES) (Hernández-Lobato et al., 2014) and Max-value Entropy Search (MES) (Wang & Jegelka, 2017), focus on reducing uncertainty about the location of the global optimum. Other uncertainty-reduction approaches, such as Expected Information Gain (EIG) (Tsilifis et al., 2017) and step-wise uncertainty reduction (Villemonteix et al., 2009), aim to clarify the black-box function itself, thereby enhancing the reliability of the surrogate model. Methods that emphasize performance gains may suffer from oversampling when the surrogate is biased or mis-specified (Wang & de Freitas, 2014), whereas uncertainty-reduction approaches do not directly target the optimizer location, potentially leading to inefficiency in identifying precise optima. To balance these two objectives, hybrid methods such as Upper Confidence Bound (UCB) (Auer et al., 2002) and Variational Entropy Search (VES) (Cheng et al., 2025) combine performance gain and uncertainty into a single reward to guide optimization. Other strategies adaptively switch between exploration and exploitation based on current observations (Bian et al., 2021).

These existing BO acquisition strategies share two common features: 1) they are typically based on human-defined subjective acquisition functions that may deviate from finding true optima: Whether they are framed in terms of the surrogate model or optimizer location uncertainty, iterative improvement with respect to specified criteria, or a combination of both, the corresponding acquisitions can be biased instead of fully aligned with the ground-truth optimizer. 2) The estimation of the acquisition functions can also introduce biases based on the chosen surrogate model space. Researchers often interpret the surrogate as a collection of candidate functions, any of which could represent the true objective function. However, this assumption may not lead to fast identification of optima in practice. For instance, the goal of BO is to identify the true optimum beyond the current best, while

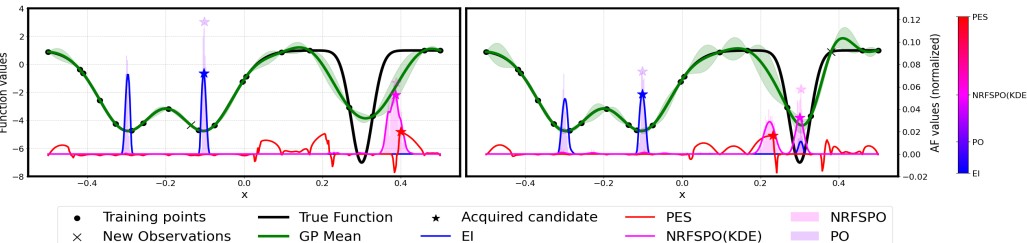

Figure 1: Acquisition behavior comparison for different acquisition strategies over two iterations: the first plot shows the initial state, and the second plot shows the state after adding the NRFS acquisition candidate to the training samples. The colormap highlights the exploration/exploitation tendencies of different acquisition strategies, where red indicates exploration and blue indicates exploitation.

during BO iterations, many sampled functions could not reach the true optimum, particularly those that already assigned the current best observation as their optimizer. This can lead to redundancy and bias towards inefficient acquisition function estimation without focusing quickly on the underlying true objective function and its optimal solution(s).

To develop a more efficient BO acquisition strategy, we focus on two key aspects: (1) what is the most suitable objective target to guide the acquisition, and (2) what is the most effective method to estimate this objective target. We develop a new probabilistic reasoning framework via Probability of Optimality (PO), which directly characterizes the likelihood of a candidate being the optimizer without relying on human-defined heuristics. Furthermore, we employ a non-replacement sampling strategy to estimate PO precisely.

Within this framework, we propose a new acquisition strategy, Non-Replacement Function Space Sampling (NRFS), which adopts a unified strategy that directly connects optimizer probability to function space coverage, consistently ensuring convergence to the global optimum. In our formulation, the surrogate model is treated as a pool of functions that contain the true objective. We iteratively identify the optimizers of functions sampled from this pool and remove these functions without replacement until the pool is fully depleted. The key advantage of this approach is that, even if the fitted surrogate is over represented in certain regions, as long as the surrogate contains the true objective, non-replacement sampling guarantees convergence once the pool is exhausted. With this strategy, no subjective reward is defined or used during the acquisition process.

Assume that we have collected observations $\mathcal{D}$, to which we fit a Gaussian process and denote the corresponding function space as $\mathcal{F}_{\mathcal{GP}}^{\mathcal{D}}$. We then take each design point $\boldsymbol{x}$ to define a *bucket*:

$$\mathcal{F}_{\boldsymbol{x}}^{\mathcal{D}} = \left\{ f \in \mathcal{F}_{\mathcal{GP}}^{\mathcal{D}} \ \middle| \ \boldsymbol{x} = \arg\min_{\boldsymbol{x}' \in \mathcal{X}} f(\boldsymbol{x}') \right\}, \tag{1}$$

which contains all functions with $\boldsymbol{x}$ being their optimizer. This definition guarantees that each function, including the objective function, belongs to at least one bucket. For functions with multiple global optima, we assign them randomly to one of their valid buckets to have one-to-one mappings between $\boldsymbol{x}$ and $\mathcal{F}_{\boldsymbol{x}}^{\mathcal{D}_n}$. If we could identify the bucket containing the true objective function, the BO task would be solved. Our approach selects the bucket containing the largest number of functions, thereby maximizing the probability that the true objective resides within the bucket. Once a bucket is selected, we remove all functions assigned to it from future consideration, ensuring that no function is selected more than once and thereby improving search efficiency.

Figure 1 illustrates a motivating example demonstrating the effectiveness of our NRFS strategy directly focusing on the global optimizer without tuning exploration-exploitation trade-off as in many existing methods. The objective is to locate the global optimum in the rightmost valley. In the initial state, EI and PO remain focused on local optima (left two valleys), where the surrogate already exhibits high confidence. This observation suggests that relying solely on PO, without incorporating NRFS, still suffers from the oversampling issue observed in EI. PES and NRFS demonstrate stronger performance on avoiding oversampling; however, after new acquisition, PES continues to target high-uncertainty areas. In contrast, NRFS shifts attention towards the promising region as shown by the pink curve in the right plot of Figure 1, enabling successful identification of the global optimum.

As we have briefly introduced the limitations of existing acquisition strategies in BO and how our proposed NRFS addresses them, we will present more details of our contribution based on the following organization: Section 2 provides a more detailed overview of commonly used acquisition functions in BO; Section 3 introduces our NRFS-based BO formulation; Section 4 reports empirical results, and Section 5 concludes the paper.

## 2 BACKGROUND

### 2.1 BAYESIAN OPTIMIZATION

Let $\boldsymbol{x}$ be a $d$-dimensional vector of decision variables in the feasible design space $\mathcal{X} \subset \mathbb{R}^d$; and $f^*(\cdot) : \mathcal{X} \to \mathbb{R}$ a continuous black-box objective function, with any evaluation $f^*(\boldsymbol{x})$ being an expensive process with respect to time and/or cost. We aim to approach the global minimizer $\boldsymbol{x}^*_{f^*}$ within a limited number of evaluations by a given function evaluation budget:

$$\boldsymbol{x}^*_{f^*} = \arg \min_{\boldsymbol{x} \in \mathcal{X}} f^*(\boldsymbol{x}). \tag{2}$$

In BO, the unknown black-box objective function is often modeled by a Gaussian process (GP), $p(f)$, characterized by its mean function $\mu(\cdot) : \mathcal{X} \to \mathbb{R}$ and covariance kernel function $k(\cdot, \cdot) : \mathcal{X}^2 \to \mathbb{R}$. BO sequentially selects a sequence of observed samples for their corresponding evaluations (Frazier, 2018). Given the observation data set until the $n$-th iteration $\mathcal{D}_n = \{\boldsymbol{X}_n, Y_n\}$, the GP posterior of $f$ is updated (Rasmussen, 2003). We denote the posterior belief of $f$ as $p(y \mid \boldsymbol{x}, \mathcal{D}_n) = P(y = f^*(\boldsymbol{x}) \mid \mathcal{D}_n)$. In each BO iteration, the next query point is chosen by optimizing an acquisition function:

$$\boldsymbol{x} = \arg \max_{\boldsymbol{x}' \in \mathcal{X}} u_n(\boldsymbol{x}'), \tag{3}$$

where $u_n(\boldsymbol{x})$ is the expected utility of evaluating $\boldsymbol{x}$ based on the updated GP posterior. The acquisition function should avoid oversampling and unnecessary exploration, which means that the resulting sequential queries should favor both the points with potential good values with respect to the objective and the informative points from the unexplored regions for learning better optimizer distribution.

### 2.2 RELATED WORK

A commonly used strategy in BO is to estimate the potential improvement obtained by evaluating a candidate point $\boldsymbol{x}$. Expected Improvement (EI) is a widely used acquisition function that accounts for potential improvements in objective value (Jones et al., 1998). A well-known limitation of EI is its tendency to oversample near local optima, particularly when the probabilistic model's prior is the initialization is biased or mis-specified (Wang & de Freitas, 2014) . Probability of Improvement (PI), which targets the likelihood of improvement, exhibits similar behavior (Kushner, 1964; Snoek et al., 2012). Oversampling usually happens when the acquisition is close to the evaluated training points, and the updated surrogate model will show minimal differences compared to the previous iteration. Consequently, the acquisition is repeatedly computed from a function set resembling the previous one, which results in nearly identical acquisition decisions iteratively.

Beyond acquisition functions that target improvement-based rewards, some strategies instead emphasize uncertainty reduction. These methods can be grouped into two categories. The first category aims to reduce uncertainty about the optimizer's location. For example, Entropy Search (ES) and PES (Villemonteix et al., 2009; Hernández-Lobato et al., 2014) explicitly model the posterior distribution over the unknown minimizer, denoted by:

$$P(\boldsymbol{x} = \boldsymbol{x}^*_{f^*} \mid \mathcal{D}_n) \approx \mathbb{E}_{f \sim \mathcal{GP}}[P(\boldsymbol{x} = \boldsymbol{x}^*_f \mid \mathcal{D}_n)] = \mathbb{E}_{f \sim \mathcal{GP}}[P(\boldsymbol{x} = \arg \min_{\boldsymbol{x}'} f(\boldsymbol{x}') \mid \mathcal{D}_n)]. \tag{4}$$

Since ES aims to reduce the uncertainty in the location of the true optimizer $\boldsymbol{x}^*_{f^*}$, the observation sequence selected is not necessarily close to $\boldsymbol{x}^*_{f^*}$, which may not provide promising candidates with optimal objective value under limited evaluation budget. In the extreme scenario where the optimizer's location is already known, PES still cannot determine the next acquisition point because the entropy contributions for optimal impossible candidates and the definitive optimizer are both 0. The second category targets reduction of function uncertainty, as in EIG or step-wise uncertainty

reduction methods, which aim to shrink the set of possible functions over the entire search space. However, this strategy can also be inefficient, since it expends extra effort distinguishing between functions that share the same optimizer.

To mitigate oversampling in a single region and suboptimal suggestions from uncertainty reduction, researchers have explored hybrid approaches, such as UCB, Moment Generating Function (MGF) (Wang et al., 2017), Truncated Variance Reduction (TVR) (Bogunovic et al., 2016) and VES, aim to balance exploration and exploitation with hyperparameters that guide acquisitions toward regions of high uncertainty when the process gets trapped in local optima. These methods typically favor either exploitation-driven or uncertainty-reduction-driven decisions in specific iterations, but rarely balance both simultaneously. Alternative hybrid methods, such as $\epsilon$-EI, enforce exploration in random iterations with a fixed probability $\epsilon$. Researchers also design schedules that decrease exploration as iterations progress. However, the optimal balance between exploration and exploitation is problem-dependent, making it challenging to predefine a universally effective schedule for diverse unknown objectives. To address this challenge, online tuning strategies are employed to dynamically adjust hyperparameters or schedules. Unfortunately, these strategies usually require at least 50 to 100 iterations to converge to optimal settings and can be even more computationally expensive for problems of high complexity.

In summary, most popular acquisition functions are driven by subjective rewards such as objective improvement, entropy reduction, variance minimization, or their combinations. While some of these strategies can yield strong empirical performance in certain cases, there still lacks a single unified policy strategy that works universally. This leads to an awkward situation in practice: identifying the most appropriate acquisition strategy often requires testing multiple options, merging existing ones, or constructing new hybrids. Rather than investing effort in developing a universal principle, the prevailing trend is to combine heuristics in the hope of achieving a better lower bound of the convergence rate, more focusing on *ad-hoc* engineering attempts by testing empirical performances. However, as long as acquisition is guided by subjective rewards, the process will remain biased towards human-designed targets rather than the true optimizer location, regardless of how much effort is spent on merging and tuning engineering heuristics.

## 3 NON-REPLACEMENT FUNCTION SPACE SAMPLING

### 3.1 OPTIMIZER PROBABILITY

To avoid redundant and/or biased BO acquisitions from subjective rewards, our new acquisition strategy, NRFS, adopts PO as our utility function, which focuses on maximizing the probability that the next acquisition corresponds to the true optimizer:

$$\boldsymbol{x} = \arg \max_{\boldsymbol{x}' \in \mathcal{X}} P(\boldsymbol{x}' = \boldsymbol{x}_{f^*}^*). \tag{5}$$

Compared to methods based on specific evaluation criteria, such as EI or PES, (5) expresses the objective of BO more directly and accurately, as it is derived from the ultimate goal in (2) and does not depend on any additional, subjectively defined rewards. However, as we do not have the underlying objective function $f^*$, the key challenge here becomes reliably estimating this probability by replacing $f^*$ in (2) with $f$ drawn from the surrogate function distribution, so that the fixed optimizer location on the left-hand side becomes a distribution over optimizer locations. The objective is then to identify the most probable optimizer location within this iteratively updated surrogate model space via this optimizer distribution.

Direct estimation of the probability $P(\boldsymbol{x} = \boldsymbol{x}_{f^*}^*)$ is nontrivial when each $\boldsymbol{x} \in \mathcal{X}$ is treated merely as an observed input location. Following the formulation strategy of Hennig & Schuler (2012), we reformulate PO as:

$$P(\boldsymbol{x} = \boldsymbol{x}_{f^*}^*) = p_{min}(\boldsymbol{x}) = \int_{f:\mathcal{X} \to \mathcal{Y}} p(f) \prod_{\substack{\tilde{\boldsymbol{x}} \in \mathcal{X} \\ \tilde{\boldsymbol{x}} \neq \boldsymbol{x}}} \theta[f(\tilde{\boldsymbol{x}}) - f(\boldsymbol{x})] \, df, \tag{6}$$

where $\theta[\cdot]$ is the Heaviside step function. The term $\prod_{\substack{\tilde{\boldsymbol{x}} \in I \\ \tilde{\boldsymbol{x}} \neq \boldsymbol{x}}} \theta[f(\tilde{\boldsymbol{x}}) - f(\boldsymbol{x})]$ acts as an indicator of whether the function $f$ regards $\boldsymbol{x}$ as its global optimizer. The above equation (6) denotes the

expected probability that $\boldsymbol{x}$ is the optimizer over all sampled functions within the surrogate model space. By combining (6) with the definition in (1), we can estimate the size of $\mathcal{F}_{\boldsymbol{x}}^{\mathcal{D}}$ as follows:

$$| \mathcal{F}_{\boldsymbol{x}}^{\mathcal{D}} |=| \mathcal{F}_{\mathcal{GP}}^{\mathcal{D}} | \int_{f:\mathcal{X}\to\mathcal{Y}} p(f) \prod_{\substack{\tilde{\boldsymbol{x}}\in\mathcal{X} \\ \tilde{\boldsymbol{x}}\neq\boldsymbol{x}}} \theta[f(\tilde{\boldsymbol{x}}) - f(\boldsymbol{x})]\, df, \tag{7}$$

where $| \mathcal{F}_{\mathcal{GP}}^{\mathcal{D}} |$ represents the cardinality of the surrogate model space. Based on (6) and (7), we establish the connection between PO and the function space coverage ratio (6), as shown:

$$P(\boldsymbol{x} = \boldsymbol{x}_{f^*}^*) = \frac{| \mathcal{F}_{\boldsymbol{x}}^{\mathcal{D}} |}{| \mathcal{F}_{\mathcal{GP}}^{\mathcal{D}} |}, \tag{8}$$

which transforms a nontrivial probability into a quantity that can be estimated by sampling functions from the surrogate GP. Note that $| \mathcal{F}_{\mathcal{GP}}^{\mathcal{D}} |$ is typically determined by how many functions are sampled from the surrogate model space, which is usually fixed. Consequently, optimizing PO is equivalent to identifying the maximizer of the numerator $| \mathcal{F}_{\boldsymbol{x}}^{\mathcal{D}} |$.

## 3.2 Coverage contribution estimation

Equation (8) suggests that increasing the probability of selecting the true optimizer can be achieved by enlarging the set of functions included in the future function bucket. However, (8) only describes a one-step strategy conditioned on some observation data $\mathcal{D}$. Since BO is inherently sequential, incorporating the acquisitions $\mathcal{D}_n$ from previous iterations is necessary. Thus, we extend (8) into its cumulative form, defined as:

$$P(\boldsymbol{x}_{f^*}^* \in \boldsymbol{X}_{n+1}) = P(\boldsymbol{x}_{f^*}^* \in \boldsymbol{X}_n) + P(\boldsymbol{x}_{n+1} = \boldsymbol{x}_{f^*}^* \mid \boldsymbol{x}_{f^*}^* \notin \boldsymbol{X}_n)P(\boldsymbol{x}_{f^*}^* \notin \boldsymbol{X}_n). \tag{9}$$

Notice that $P(\boldsymbol{x}_{f^*}^* \in \boldsymbol{X}_n)$ and $P(\boldsymbol{x}_{f^*}^* \notin \boldsymbol{X}_n)$ are fixed after $n$ iterations, optimization of (9) can be achieved by maximization on term $P(\boldsymbol{x}_{n+1} = \boldsymbol{x}_{f^*}^* \mid \boldsymbol{x}_{f^*}^* \notin \boldsymbol{X}_n)$. Unlike the one-step maximization target defined in (8), $P(\boldsymbol{x}_{n+1} = \boldsymbol{x}_{f^*}^* \mid \boldsymbol{x}_{f^*}^* \notin \boldsymbol{X}_n)$ is conditioned on the event $\boldsymbol{x}_{f^*}^* \notin \boldsymbol{X}_n$. Intuitively, this is reasonable: if $\boldsymbol{x}_{f^*}^* \in \boldsymbol{X}_n$, the BO process would already be completed. Thus, (9) formalizes that continuing BO requires conditioning on $\boldsymbol{x}_{f^*}^* \notin \boldsymbol{X}_n$.

To the best of our knowledge, no prior work has used $\boldsymbol{x}_{f^*}^* \notin \boldsymbol{X}_n$ as a condition to guide BO, since this condition appears to merely shrink the design space. However, we observe that $\boldsymbol{x}_{f^*}^* \notin \boldsymbol{X}_n$ also influences the objective space, thereby exerting a broader impact on the function space. From $\boldsymbol{x}_{f^*}^* \notin \boldsymbol{X}_n$, we obtain a corresponding condition in function space: $f^*(\boldsymbol{x}_{f^*}^*) \notin \boldsymbol{Y}_n$. For a minimization problem, this is equivalent to $f^*(\boldsymbol{x}_{f^*}^*) < \boldsymbol{Y}_n^* = \min\{\boldsymbol{Y}_n\}$. This inequality condition is crucial, as it indicates that the true optimum must improve upon the current best. Consequently, any sampled function that cannot achieve a value better than the current best cannot be the true objective function. When sampling the function space to estimate (9), such functions should be excluded, as their probability $p(f)$ is zero.

Thus to maximize (9), we must consider how many functions from $\mathcal{F}_{\boldsymbol{x}}^{\mathcal{D}}$ are from the objective function impossible region and remove them from future consideration. We partition the candidate's function cluster into two subsets:

$$\mathcal{F}_{\boldsymbol{x}}^{D_n} = \mathcal{F}_{\boldsymbol{x},f(\boldsymbol{x})\geq \boldsymbol{Y}_n^*}^{D_n} \cup \mathcal{F}_{\boldsymbol{x},f(\boldsymbol{x})<\boldsymbol{Y}_n^*}^{D_n} \tag{10}$$

For all functions in $\mathcal{F}_{\boldsymbol{x},f(\boldsymbol{x})\geq \boldsymbol{Y}_n^*}^{\mathcal{D}_n}$, their optima do not surpass the current best observation $\boldsymbol{Y}_n^*$. As discussed earlier, the true objective function cannot belong to this set. Consequently, functions from set $\mathcal{F}_{\boldsymbol{x},f(\boldsymbol{x})\geq \boldsymbol{Y}_n^*}^{\mathcal{D}_n}$ contribute only ineffective coverage. Therefore, the effective coverage can be computed as:

$$|\mathcal{F}_{\boldsymbol{x}}^{D_n}| = |\mathcal{F}_{\boldsymbol{x},f(\boldsymbol{x})<\boldsymbol{Y}_n^*}^{D_n}| + |\mathcal{F}_{\boldsymbol{x},f(\boldsymbol{x})\geq \boldsymbol{Y}_n^*}^{D_n}| = |\mathcal{F}_{\boldsymbol{x},f(\boldsymbol{x})<\boldsymbol{Y}_n^*}^{D_n}|. \tag{11}$$

To accurately estimate the coverage contribution of each candidate point $\boldsymbol{x}$, we restrict sampling optimizers that overwhelm the current best for all $\boldsymbol{x} \in \mathcal{X}$. This constraint ensures unbiased estimation of coverage improvement across all possible candidates. Under this setup we sample from a Truncated Gaussian Process (TGP), defined as $\mathcal{TGP}(\mu, k, t)$, where $t$ is the threshold. For minimization problems, we sample from $\mathcal{TGP}^-(\mu, k, \boldsymbol{Y}_n^*)$, indicating that the sampling is restricted to

the function space lying below the threshold $\boldsymbol{Y}_n^*$. The final utility function is estimated by a variant of (6) with the samples from $\mathcal{TGP}^-(\mu, k, \boldsymbol{Y}_n^*)$, shown as:

$$\mathbb{E}_{f \sim \mathcal{TGP}^-(\mu,k,\boldsymbol{Y}_n^*)} \left[ P\left(\boldsymbol{x} = \boldsymbol{x}_f^* \mid \mathcal{D}_n\right)\right] = \int_{-\infty}^{\boldsymbol{Y}_n^*} \prod_{\substack{\tilde{\boldsymbol{x}} \in \mathcal{X} \\ \tilde{\boldsymbol{x}} \notin \{\boldsymbol{X}_n \cup \boldsymbol{x}\}}} \theta[f(\tilde{\boldsymbol{x}}) - f(\boldsymbol{x})]p(f)df. \quad (12)$$

In summary, we maximize (9) by identifying the maximizer of (12). From (12), we can illustrate how NRFS avoids the exploration–exploitation dilemma based on joint distribution of probability of optimality and function distribution. Compared with the original probability of optimality without any conditioning, our method by NRFS mitigates oversampling by filtering out all functions that have been identified with an optimizer whose $p(f) = 0$ in future steps. Meanwhile, the threshold condition $\boldsymbol{Y}_n^*$ also prevents redundancy by acquisitions for uncertainty reduction in regions where the optimizer cannot exist, thereby enhancing acquisition efficiency.

### 3.3 ONE-STEP-LOOK-AHEAD VARIANT DEVELOPMENT

To further increase sample efficiency in BO, here we develop a one-step-look-ahead(OSLA) variant of (12). After $T$ evaluations, the probability that the optimizer has been identified is

$$R_T = P(\boldsymbol{x}_{f^*}^* \in \boldsymbol{X}_T) = 1 - \prod_{t=1}^{T}\left(1 - P(\boldsymbol{x}_{f^*}^* = \boldsymbol{x}_t)\right). \quad (13)$$

We refer to $R_T$ as the cumulative success probability. The corresponding incremental contribution of iteration $t$ is

$$r_t = P(\boldsymbol{x}_{f^*}^* = \boldsymbol{x}_t) \prod_{s=1}^{t-1}\left(1 - P(\boldsymbol{x}_{f^*}^* = \boldsymbol{x}_s)\right), \quad (14)$$

i.e., the probability of discovering the optimizer at iteration $t$ given it has not been selected before. At iteration $n$, maximizing the defined cumulative success probability is equivalent to maximizing the following one-step-look-ahead value function:

$$V_n \approx r_n + r_{n+1} = \prod_{s=1}^{n-1}\left(1 - P(\boldsymbol{x}_{f^*}^* = \boldsymbol{x}_s)\right)\left[P(\boldsymbol{x}_{f^*}^* = \boldsymbol{x}_n) + \left(1 - P(\boldsymbol{x}_{f^*}^* = \boldsymbol{x}_n)\right)P(\boldsymbol{x}_{f^*}^* = \boldsymbol{x}_{n+1})\right]. \quad (15)$$

Since $\prod_{s=1}^{n-1}\left(1 - P(\boldsymbol{x}_{f^*}^* = \boldsymbol{x}_s)\right)$ is fixed after iteration $n-1$, optimization of the value function reduces to maximizing $P(\boldsymbol{x}_{f^*}^* = \boldsymbol{x}_n) + \left(1 - P(\boldsymbol{x}_{f^*}^* = \boldsymbol{x}_n)\right)P(\boldsymbol{x}_{f^*}^* = \boldsymbol{x}_{n+1})$. We therefore define the corresponding one-step-look-ahead utility function as:

$$u_n(\boldsymbol{x}) = P(\boldsymbol{x}_{f^*}^* = \boldsymbol{x} \mid \mathcal{D}_n) + \left(1 - P(\boldsymbol{x}_{f^*}^* = \boldsymbol{x} \mid \mathcal{D}_n)\right)\mathbb{E}_{y \mid \mathcal{D}_n, \boldsymbol{x}}\left[\max_{\boldsymbol{x}' \in \mathcal{X}} P\left(\boldsymbol{x}_{f^*}^* = \boldsymbol{x}' \mid \mathcal{D}_n \cup (\boldsymbol{x}, y)\right)\right], \quad (16)$$

where $P(\boldsymbol{x}_{f^*}^* = \boldsymbol{x} \mid \mathcal{D}_n)$ and $\max_{\boldsymbol{x}' \in \mathcal{X}} P\left(\boldsymbol{x}_{f^*}^* = \boldsymbol{x}' \mid \mathcal{D}_n \cup (\boldsymbol{x}, y)\right)$ are estimated by (12).

In contrast to one-step-look-ahead variants of other acquisition functions, which typically require tuning a hyperparameter $\gamma$ to balance immediate and future rewards, our one-step-look-ahead NRFS directly sets $\gamma = 1 - P(\boldsymbol{x}_{f^*}^* = \boldsymbol{x} \mid \mathcal{D}_n)$. This choice avoids potential probability gaps across different problem settings, eliminating the need to search for task-dependent hyperparameters.

Another advantage is that the one-step-look-ahead NRFS variant has the potential to achieve the maximum convergence rate. Traditionally, the convergence rate is defined by how quickly an error term, or a probability gap, decays to zero (Ryzhov, 2016; Bull, 2011). In our setting, this probability gap corresponds to the cumulative regret $R_T^{\text{reg}} = 1 - R_T$. Note that maximizing the value function $V_n$ at each step is equivalent to minimizing cumulative regret, so each acquisition step can be interpreted as directly advancing the convergence rate.

## 4 EMPIRICAL RESULTS

### 4.1 EXPERIMENTAL SETUP

We follow the practice in PES (Hernández-Lobato et al., 2014), to estimate (12) and (16). Specifically, we first sample $M$ (1000) functions from $\mathcal{TGP}^-(\mu, k, \boldsymbol{Y}_n^*)$ to obtain a function set $\{f_j(\cdot)\}_{j=1}^M$.

When $M = 1$, the approach degrades to a variant of Thompson sampling (Russo et al., 2018), but with samples drawn from a TGP instead of the full surrogate. For each sampled function $f_j(\cdot)$, we identify its optimizer location and then aggregate these optimizers to estimate the distribution over the design space. In continuous domains, NRFS can be implemented by applying density estimation techniques such as Kernel Density Estimation (KDE) (Chen, 2017), Gaussian Mixture Models (GMM), $k$-Nearest Neighbors (KNN), or any other strategies to approximate the optimizer distribution. In our implementation, we apply Parzen estimator (Silverman, 2018), resulting in a continuous density function. The acquisition is selected as the most probable optimizer location under this estimated distribution.

## 4.2 SEARCHING BEHAVIOR

To understand the search behavior of NRFS intuitively, we test an illustrative example with a Gaussian mixture (GM) objective function to minimize: $-\mathcal{N}(-\mu_1, \sigma_1^2) - \mathcal{N}(\mu_2, \sigma_1^2) - 0.55\mathcal{N}(\mu_1, \sigma_2^2) + 1$. We set $\mu_1 = 0.3$, $\mu_2 = -0.1$, $\sigma_1 = 0.05$ and $\sigma_2 = \frac{0.3}{\sqrt{2}}$ (Figure 1). The objective function has two local minima and one global minimum far from local minima. The searching behavior comparison detailed discussion can be found in Section 1 and more empirical performance comparison can be found in Section 4.3.

Besides the search behavior analysis on 1D GM example, we apply NRFS on other benchmark objective functions which are more commonly used in evaluating BO methods, including 1) Branin (Branin, 1972); 2) BraninRcos2 (Al-Roomi, 2015); 3) Himmelblau's (Himmelblau et al., 2018) and 4) Forrester2D (Forrester et al., 2008) to check the convergence. Figure 2 demonstrates that, given a sufficient number of iterations, NRFS converges to all global optima. The distribution of training samples shows high density near the true optima and lower density elsewhere, indicating that acquisition is guided by the potential to identify global optima in given regions. Oversampling occurs rarely and primarily when the search is already close to an optimum. At the same time, later iterations still allocate some acquisitions in high uncertainty regions, which avoids missing possible global optimum which has extreme objective value. Additional behavioral analysis on other objective functions is provided in Section A.1 of the *Appendix*.

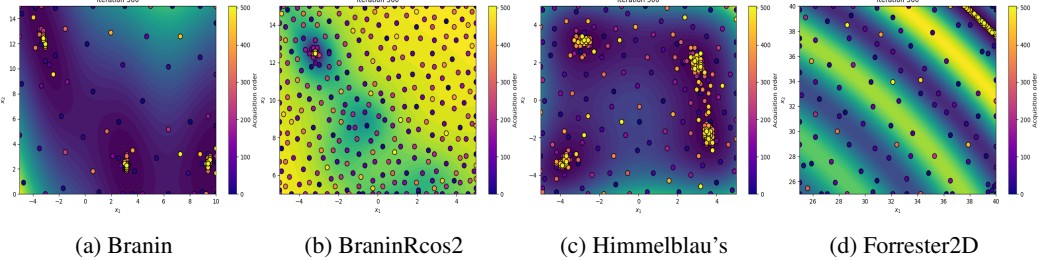

| (a) Branin | (b) BraninRcos2 | (c) Himmelblau's | (d) Forrester2D |

Figure 2: Convergence performance on different objective functions with multiple local optima: (a) Branin; (b) BraninRcos2; (c) Himmelblau's; and (d) Forrester2D. Darker regions indicate lower values, which are more desirable for minimization task. White dots mark the iterative acquisitions.

## 4.3 ACQUISITION FUNCTION EVALUATION

We compare NRFS and its one-step-look-ahead variant against several baselines including EI (Jones et al., 1998), PES (Villemonteix et al., 2009; Hernández-Lobato et al., 2014), $\epsilon$-EI, UCB (Srinivas et al., 2009), Tree-structured Parzen Estimator (TPE) (Bergstra et al., 2011; Watanabe, 2023), PI (Kushner, 1964; Snoek et al., 2012), Knowledge Gradient (KG) (Frazier et al., 2008), and Latin Hypercube random sampling (RS) (McKay et al., 2000). EI, $\epsilon$-EI, PI, UCB, PES, KG are implemented within the BOTorch framework (Balandat et al., 2020), while TPE is based on Optuna (Akiba et al., 2019). More discussion of these previous acquisition function is in *Appendix* A.2.

We have evaluated all acquisition functions on four known objective functions: the GM function introduced in Section 4.2, the 2D Forrester function (Forrester et al., 2008), a modified Rosenbrock function (Al-Roomi, 2015; Rosenbrock, 1960), and the Shekel function (Molga & Smutnicki, 2005).

These known objective functions are selected because they require a balance of exploration and exploitation to effectively locate the global optimum. Visualizations of the objective landscapes and analyses of search behaviors under different BO strategies are provided in Section A.1 of the *Appendix*. These four functions represent three types of multi-modal objective landscapes: 1) The GM and modified Rosenbrock functions exhibit large, smooth local optima along with a sharp global optimum distant to local optima; 2) The 2D Forrester function contains large, smooth local optima and a similarly smooth global optimum; 3) The Shekel function features multiple sharp peaks, with all optima being narrow and distinct. In addition to the four known objective functions, we consider two real-world case studies aimed at identifying the optimal composition of six elemental materials (Hastings et al., 2025). The first case targets the highest stacking fault energy (SFE), while the second focuses on maximizing heat capacity (HC). In both cases, the mapping from composition to property is unknown.

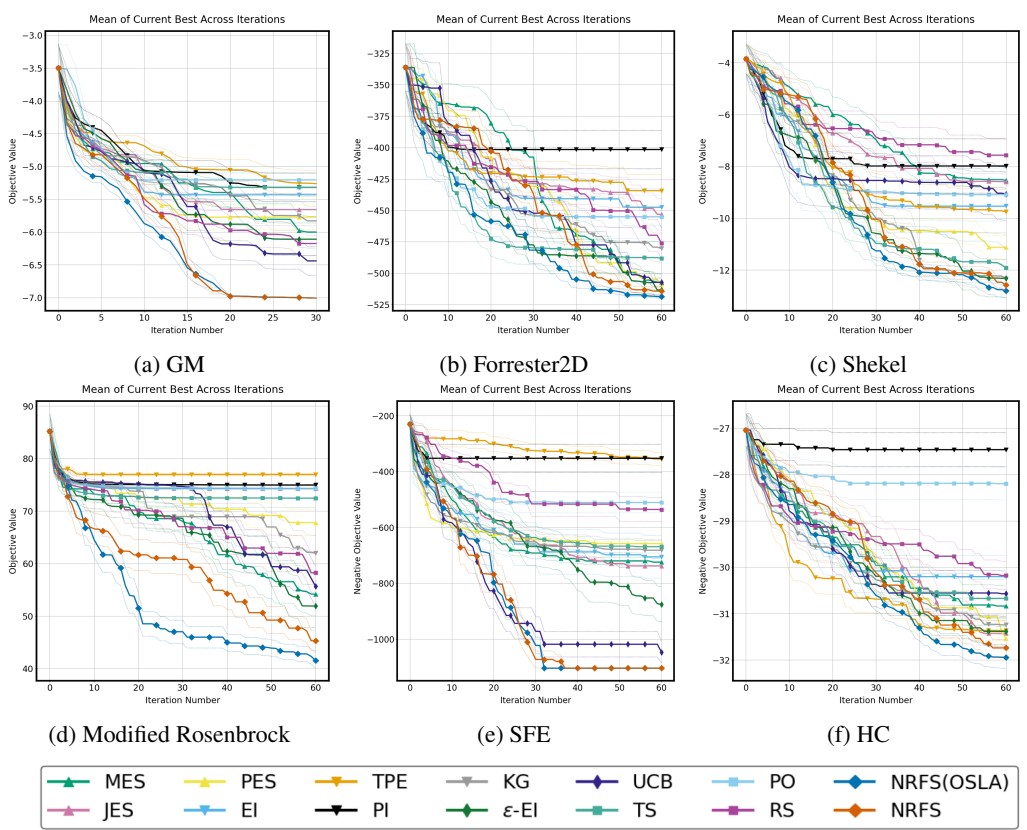

Figure 3: Performance comparison of 20 independent trials on different objective functions: (a) 2D Gaussian mixture model; (b) Forrester2D; (c) Shekel; (d) Modified Rosenbrock; (e) SFE and (f) HC

In the cases of GM, the modified Rosenbrock function and SFE, our NRFS consistently outperforms all other competing methods, not only consistently achieving better objective function values across iterations but also exhibiting greater stability. These objective functions require BO to escape local optima and effectively exploit the global optimal region to achieve faster convergence. In the Forrester2D case, $\epsilon$-EI and UCB also demonstrate competitive performance compared to NRFS and one-step-look-ahead NRFS. This is because the global optimal region is relatively large and smooth, unlike previous three examples. As a result, $\epsilon$-EI and UCB can more easily acquire points in the global optimal region via random sampling or uncertainty reduction. However, achieving the performance shown in Figure 3b requires sweeping $\epsilon$ from 0.1 to 0.9 to identify the best-performing hyperparameter, resulting in significantly higher evaluation cost compared to NRFS and one-step-look-ahead NRFS. For UCB, even when using a self-adjusting schedule $\beta = 0.5d \cdot \log(n + 1) \cdot c$, selecting the constant $c$ remains challenging due to the lack of prior knowledge about the scale and smoothness of the objective function, making it difficult to define a reasonable hyperparameter range. In contrast, our NRFS-based BO methods do not require hyperparameter tuning compared

to these methods. In the Shekel and HC cases, NRFS-based methods are initially outperformed by other acquisition functions in the early iterations. However, as the number of BO iterations increases, they achieve the best overall performance. The strong early performance of value improvement based methods can be attributed to their rapid convergence toward local minima. Nevertheless, as the optimization progresses, these methods eventually fail to provide meaningful acquisitions due to their inherently unbalanced acquisition strategies.

More importantly, the landscape of the objective function is unknown before evaluation. Consequently, our NRFS-based methods offer the most robust BO strategy among all the alternatives, as they consistently demonstrate stable performance improvements and reliably converge to the global optimum regardless the landscape of the objective function. A clearer presentation of their standard deviation (std) performance in Section A.3 of the *Appendix* further highlights their reliability.

In addition to the sequential NRFS strategy, we also evaluate a batch variant NRFS on the four synthetic objective functions. For batch selection, we continue to follow a non-replacement strategy: once the candidate with the highest probability of being the optimizer (per 12) is identified, we remove all sampled function realizations that designate this candidate as their optimizer. From the remaining realizations, we then select the next most probable optimizer location. This process is repeated until we obtain a full batch of acquisition points. To ensure fairness, we fix the evaluation budget to be identical to that of the sequential acquisition and assess how varying the batch size impacts optimization performance under the same evaluation limit.

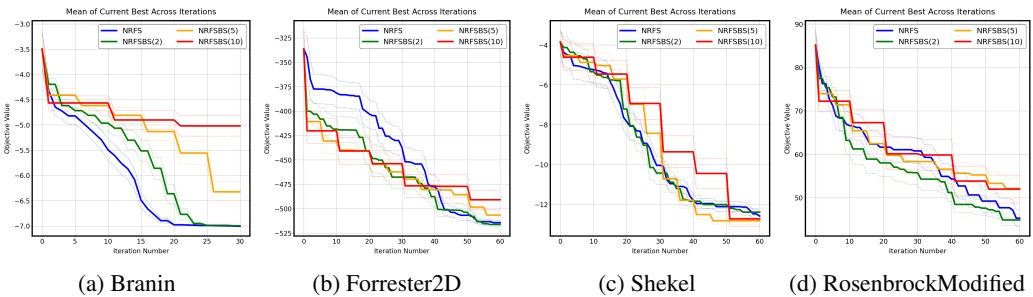

| (a) Branin | (b) Forrester2D | (c) Shekel | (d) RosenbrockModified |

Figure 4: Batch performance comparison of 20 independent trials on different objective functions: (a) 2D Gaussian mixture model; (b) Forrester2D; (c) Shekel and (d) Modified Rosenbrock

As shown in Figure 4, sequential optimization usually has the best performance. The only exception is the Shekel function, where large batch methods perform better. This difference comes from the structure of the Shekel objective: the global optimum is located in a very sharp and narrow region. Once this region is identified, batch acquisition functions tend to place many points directly around that area because the optimizer distribution becomes highly concentrated there, allowing batches to exploit the optimum quickly. In contrast, sequential acquisition updates the surrogate and the current best after each evaluation, which smooths the optimizer distribution and reduces the likelihood of repeatedly sampling the same small region.

To further assess NRFS characteristics, we extend our study to high dimensional variants of the Forrester function. We benchmark the same 14 acquisition strategies used in Figure 3 across 5D, 10D, 20D, and 50D. The initialization number ($5d$) is adjusted according to dimensionality to ensure optimization efficiency. As Figure 5 shows, the performance gap between acquisition strategies shrinks as the dimensionality increases. By the time the dimension reaches 50, all methods struggle to discover substantially better objective values. Nevertheless, NRFS and NRFS(OSLA) maintain superior performance compared to the other strategies. We have also performed benchmarking on other commonly used objective functions, real case applications and robustness test against noise with different levels for all of the acquisition methods mentioned above. The results could be found in Sections A.4 and A.5 of the *Appendix*. Section A.6 includes the complexity analysis and computational cost comparison for different acquisition methods.

Beyond empirical performance comparison, we further analyze the acquisition behaviors of EI, PES, and NRFS on the SFE task, highlighting how NRFS can facilitate real-world material discovery. The top $5\%$ performing materials are distributed across three regions, with only one region containing the global optimizer. We compare the performance of our NRFS with the BO methods using EI

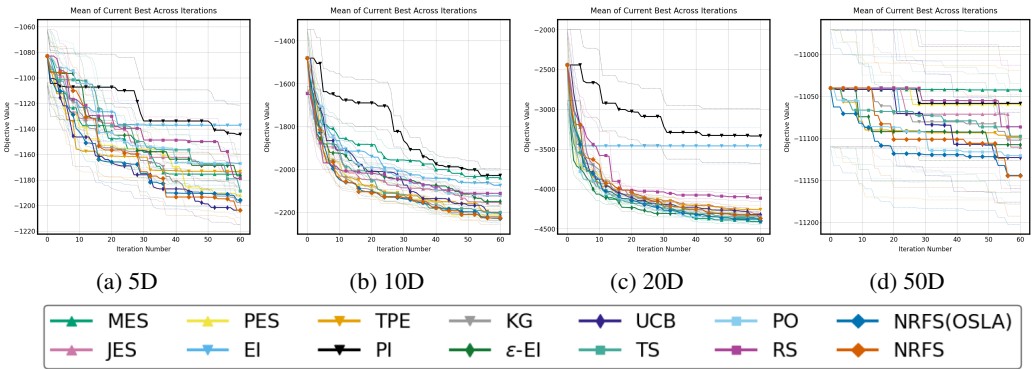

Figure 5: Optimization performance comparison of 20 independent trials on Forrester functions with different dimensions: (a) 5D; (b) 10D; (c) 20D and (d) 50D

and PES acquisition functions. Figure 6 illustrates the BO performance difference, where NRFS successfully finds the global minimum but EI and PES fail within 40 iterations. As illustrated, EI gets trapped in the local minimum; PES fails to identify the exact global minimum after getting close, suffering from its focus on information gain instead of acquisition in potential optimal region.

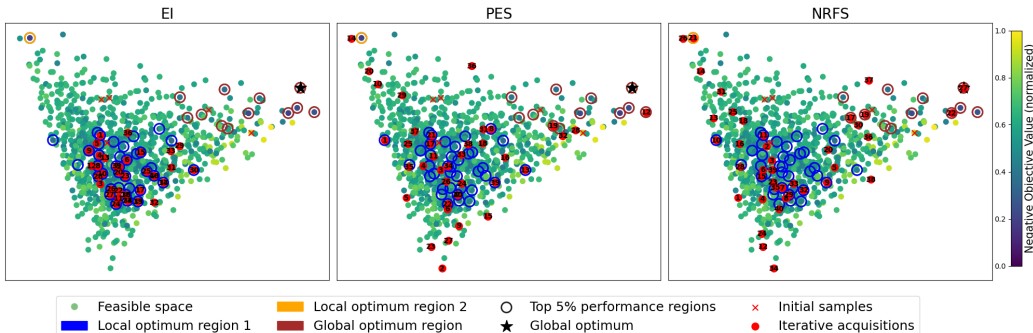

Figure 6: Acquisition behavior comparison of different strategies on the SFE material rediscovery task over 40 iterations. Darker candidates correspond to higher SFE values. Global optimizer is marked with a black star. Black numbers in red disks represent the iteration of acquisitions by three different strategies, among which only NRFS reaches the global optimizer at iteration 27 while EI and PES fails to identify it within 40 iterations.

## 5 CONCLUSION & FUTURE RESEARCH

We have proposed a novel BO strategy, NRFS, which acquires the query candidate that has the maximum probability to be the true optimizer. To further enhance acquisition efficiency, we transform the estimation of this probability to estimating function space coverage by focusing on surrogate functions whose optima are likely to be the true optimum. We provide strong empirical evidence that NRFS can converge to all global optima for a diverse family of benchmark objective functions and demonstrate superior empirical advantages of NRFS over existing BO baselines. Moreover, NRFS not only provides a new BO strategy, but also opens a new research direction on how one should utilize the surrogate models more efficiently. The computation of other acquisition functions in other BO methods can also be based on the sampled functions from NRFS.

A natural extension of NRFS is to multi-objective optimization, where truncation is defined by the Pareto front rather than a single threshold, making sampling substantially more challenging. Another promising direction is improving NRFS's computational efficiency through analytical formulations, which could also enable explicit convergence-rate guarantees and strengthen its theoretical foundation.

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

# A APPENDIX

## A.1 SEARCHING BEHAVIORS

In addition to the search behavior analysis on the illustrative Gaussian Mixture (GM) example in Section 4.2 and stacking fault energy (SFE) real case presented in Section 4.3 of the main text, we provide additional analyses based on the BO results on different objective functions discussed in Section 4.3. This extended comparison highlights the differences in search behavior dynamics among commonly used non-hybrid acquisition strategies: Expected Improvement (EI) (Jones et al., 1998), Predictive Entropy Search (PES) (Hernández-Lobato et al., 2014), and our proposed NRFS.

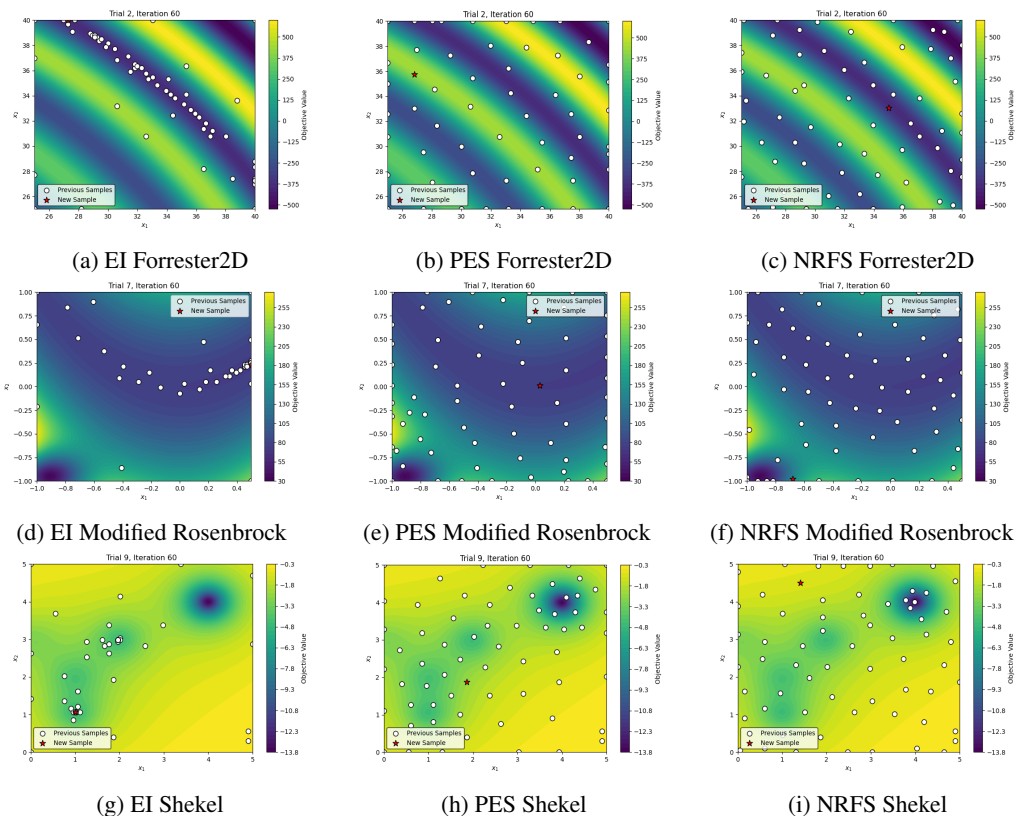

(a) EI Forrester2D      (b) PES Forrester2D      (c) NRFS Forrester2D

(d) EI Modified Rosenbrock      (e) PES Modified Rosenbrock      (f) NRFS Modified Rosenbrock

(g) EI Shekel      (h) PES Shekel      (i) NRFS Shekel

Figure 7: Search behavior comparison between EI-, PES- and NRFS-based BO. Red stars: Next acquisitions; White dots: Previous acquisitions.

For the three minimization cases in Figure 7 above, EI tends to oversample regions near local optima, highly likely to continue until those regions are thoroughly exploited. In contrast, PES exhibits the opposite behavior: even after identifying a candidate in a region likely to contain the global

optimum, it continues to explore points that reduce uncertainty about the optimizer's location, rather than concentrating on the regions with high potential. NRFS demonstrates a more balanced search strategy. Its acquisition distribution is closely aligned with the objective landscape, assigning higher acquisition frequency to regions closer to the optimum.

## A.2 BASELINE METHODS DISCUSSION

In this section, we review the existing acquisition functions and explain why they are used as our comparison baselines. We start with EI. As illustrated in Fig. 1, EI and PO exhibit similar performance, both being affected by the oversampling issue. Many prior studies (Hennig & Schuler, 2012) also use EI for GP global PO estimation, however, the underlying principles of PO and EI are fundamentally different. Consider a simple case with two candidate buckets. While most functions identify the first bucket as containing the optimizer, its expected improvement is lower than that of the second bucket because the second buckets contains optimum with extreme value. As a result, EI selects the second bucket due to the higher expected improvement. However, from an intuitive perspective, the first bucket should be preferred since it has a higher likelihood of containing the true objective function. This misstep arises because EI prioritizes potential magnitude over the likelihood of containing the true optimum which is an inherent property of the objective that remains fixed regardless of human-defined expectations. The difference between EI and PO in Fig. 1 is not pronounced because the objective values follow a Gaussian distribution with small variance, different from the earlier examples with those following extreme-value distributions. When the Gaussian distribution has larger variance their difference becomes more apparent, especially in the design space boundary region (Hennig & Schuler, 2012).

For uncertainty reduction based methods, PES aims to reduce the number of remaining candidate buckets rather than directly selecting the most probable one. Methods with function uncertainty reduction ability like UCB, TVR and Expected Information Gain (EIG) (Tsilifis et al., 2017), primarily focus on decreasing the total number of possible functions across all buckets. This approach can be inefficient, as it devotes unnecessary effort to keep reducing the number of functions including the buckets that have already been considered.

Besides the most popular acquisition functions such as EI, PES and UCB. TPE is a variant of EI, and we include it in our comparison because it also incorporates both truncated modeling and density estimation over the design space. Given a minimization problem, TPE sets the threshold above the current best value, and uses candidate locations with objective values below this threshold to estimate the optimizer density. Our goal is to evaluate how its truncation strategy compares to ours. KG, a popular one-step-look-ahead strategy, that estimates the expected value of information from evaluating a candidate point, which we include as a baseline to compare with our one-step-look-ahead NRFS. PI, a replacement function space sampling strategy, that focuses solely on the likelihood of improving over the current best observation rather than targeting global optimality, is used as the baseline to compare the difference between Non-replacement and replacement sampling, objective improvement and optimality. RS is included as a non-informative baseline to assess optimization performance.

## A.3 PERFORMANCE CONSISTENCY

We further extract the standard deviation values to illustrate the performance consistency explicitly in this section.

We observe that, across all six case studies with different objective functions as discussed in the main text, our NRFS exhibits the most stable performance among the cases with GM, Forrester2D and Shekel. For Modified Rosenbrock, as Fig. 8d shows, EI, TPE, and PI demonstrate more consistent behavior than NRFS, likely due to their tendency to oversample in local optima across all 20 trials. By combining the mean performance reported in the main text with the observed standard deviation values, we find that NRFS exhibits low variability only when it converges to the global optimum. This behavior suggests that std can serve as a reliable indicator of proximity to the global optimum: since NRFS does not suffer from oversampling, a near-zero std implies that the current best is close to the global optimum.

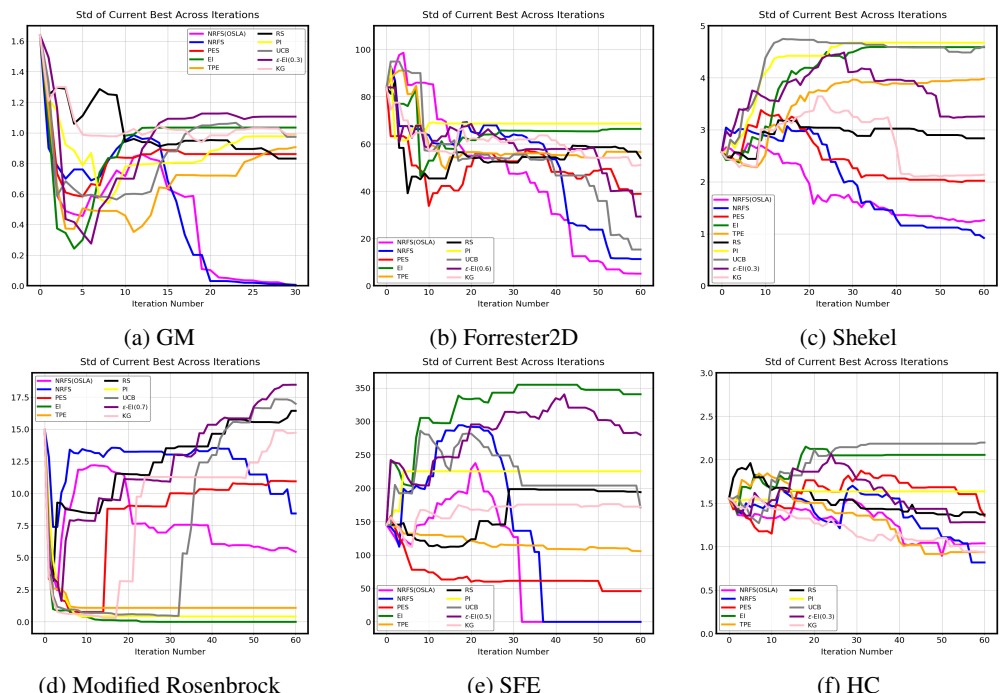

(a) GM      (b) Forrester2D      (c) Shekel

(d) Modified Rosenbrock      (e) SFE      (f) HC

Figure 8: Std performance comparison on different objective functions: (a) 2D Gaussian mixture model; (b) Forrester2D; (c) Shekel; (d) Modified Rosenbrock; (e) SFE and (f) HC

## A.4 ADDITIONAL BENCHMARKING RESULTS

Besides the objective functions that require a balanced strategy, we also consider Branin (Dixon, 1978) as a purely exploitation-driven case, and BraninRcos2 (Al-Roomi, 2015) as an exploration-driven case, to evaluate NRFS under these extremes. Branin contains three global optima, and identifying any one of them suffices to achieve optimal performance. In contrast, BraninRcos2 resembles the Shekel function, characterized by many local optima and a relatively narrower global optimum region. This implies that once the global optimum region is explored, the optimal solution is nearly recovered.

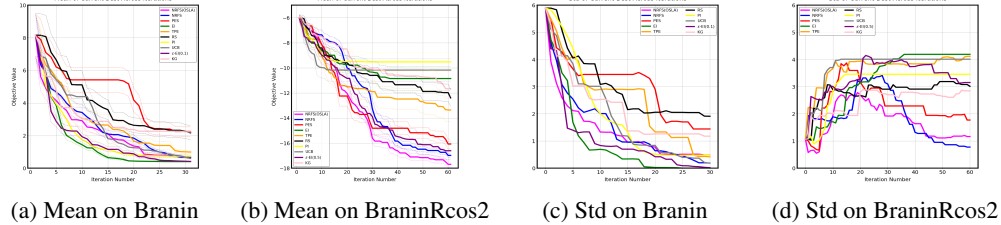

(a) Mean on Branin      (b) Mean on BraninRcos2      (c) Std on Branin      (d) Std on BraninRcos2

Figure 9: Performance comparison of BO methods on 2 extra objectives: (a, b) current best of 20 trials for Branin and BraninRcos2; (d, e) std of the current best on the same benchmarks.

As Figure 10 shows, in the Branin case, NRFS appears to require more iterations to converge to the global optimum compared to EI. As illustrated in Figure 2a (Section 4.2), this slower convergence can be interpreted as the inherent trade-off for acquiring all global optima rather than focusing on a single one. This observation indicates that a current limitation of NRFS: it may not outperform some existing methods when the problem contains only one optimal region or multiple optimal regions with the same optimal values. In such cases, identifying one region is sufficient to guarantee the global optimum, thus there is no need to acquire candidates from other high potential regions. In the BraninRcos2 setting, the performance trend aligns with that observed in the Shekel function (Figure 3c, Section 4.3). Specifically, PES is able to identify the global optimum region more quickly

810 in the initial iterations. However, once the optimal region is located, NRFS starts to outperform PES
811 in subsequent acquisitions.

813 We further evaluate our method on three real-world benchmark tasks: (1) a 124-dimensional soft-
814 constrained variant of the Mopta08 automotive design problem, originally introduced by (Jones,
815 2008); (2) the 180-dimensional Lasso-DNA objective selection task, proposed by (Šehić et al.,
816 2022), which involves optimizing hyperparameters of a sparse regression model used to recover
817 underlying genetic signal patterns; and (3) a 60-dimensional rover trajectory-planning benchmark
818 from (Wang et al., 2018), where the optimizer must choose spline-parameterized way points to min-
819 imize traversal cost in the presence of obstacles.

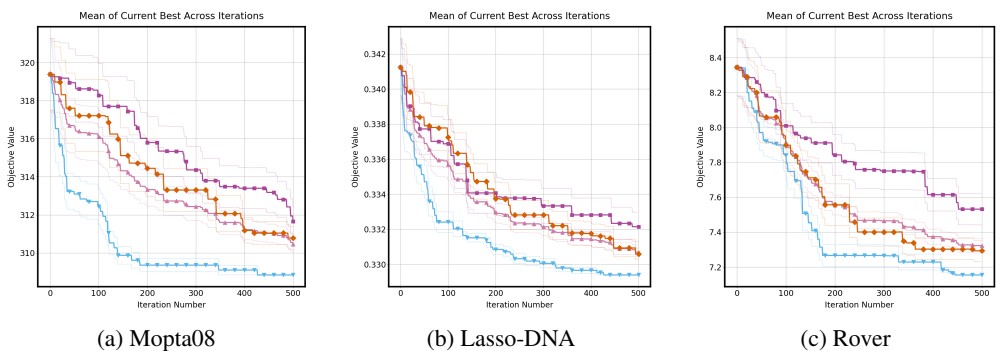

(a) Mopta08      (b) Lasso-DNA      (c) Rover

Figure 10: Performance comparison of BO methods on three objectives: (a, b) current best of 20
trials for Branin and BraninRcos2; (d, e) std of the current best on the same benchmarks.

## A.5 ROBUSTNESS ANALYSIS

We have also evaluated robustness by adding varying levels of Gaussian noise to the four objec-
tive functions described in the main text. For hybrid acquisition strategies such as UCB and $\epsilon$-EI,
hyperparameters were further tuned to ensure optimal performance under different noise levels.

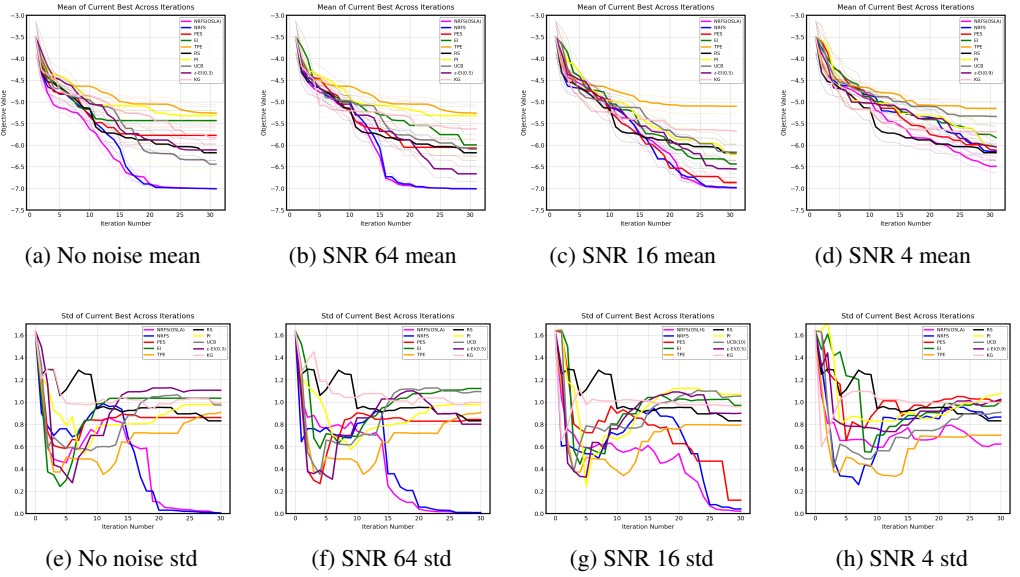

(a) No noise mean    (b) SNR 64 mean    (c) SNR 16 mean    (d) SNR 4 mean

(e) No noise std    (f) SNR 64 std    (g) SNR 16 std    (h) SNR 4 std

Figure 11: Performance comparison for the GM example with different signal-to-noise ratio (SNR)
levels.

Based on the results, we conclude that NRFS is robust to different noise levels corresponding to
SNR values above 16 across all four objective functions. It even demonstrates robustness with high

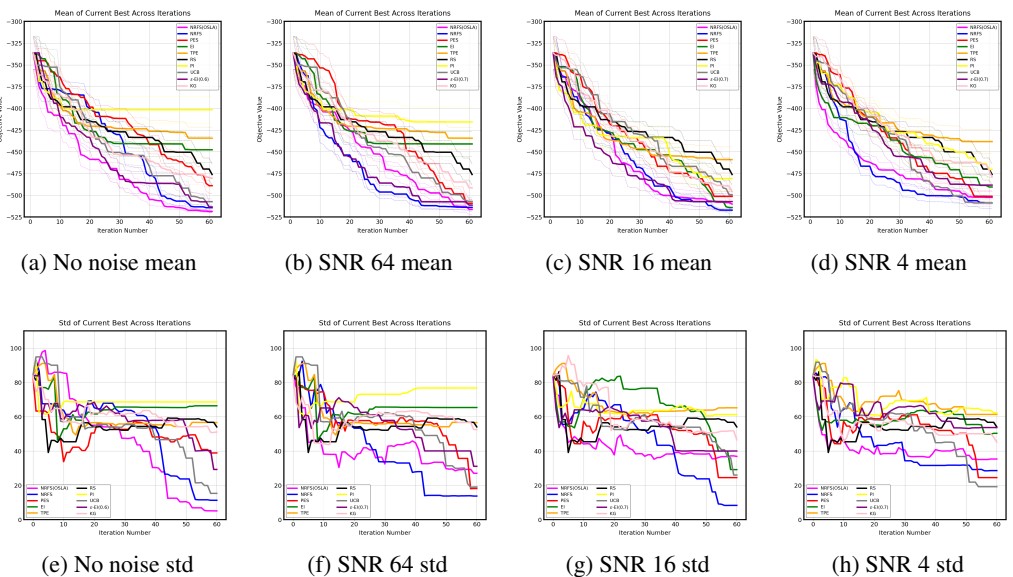

Figure 12: Performance comparison on Forrester2D function with different SNR levels.

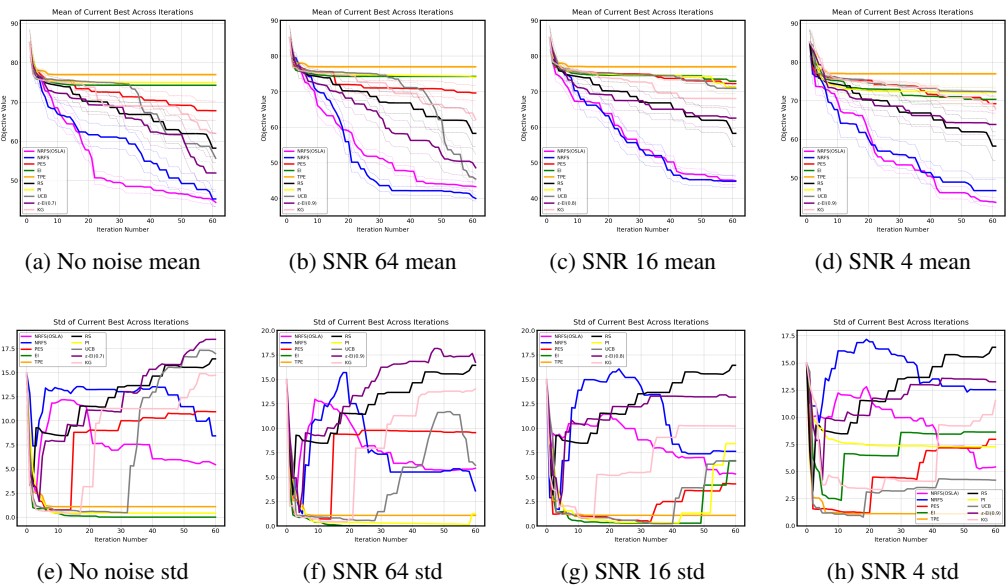

Figure 13: Performance comparison on modified Rosenbrock function with different SNR levels.

noise (SNR = 4) in the Modified Rosenbrock and Forrester2D cases. For NRFS, adding noise to the surrogate model broadens the effective function space, which is not necessarily detrimental. In fact, as shown in Figures. 12b and 13b, small amount of noise can improve performance compared to the noise-free cases. The primary negative impact of noise arises from inaccurate estimation of the previous best values. When the noise level is high, overestimation of the incumbent solution may cause the true optimum to be bypassed, and our non-replacement sampling strategy may consequently skip the global optimum.

While one-step-look-ahead NRFS is occasionally outperformed by standard NRFS, as shown in Figures. 12b, 12d and 13b, this degradation in performance can be attributed to the fact that the one-step-look-ahead strategy may increase the risk of overestimating the current best.

Among the remaining methods, TPE demonstrates the most stable performance, showing no significant degradation or improvement across all four objective functions under varying noise levels. This

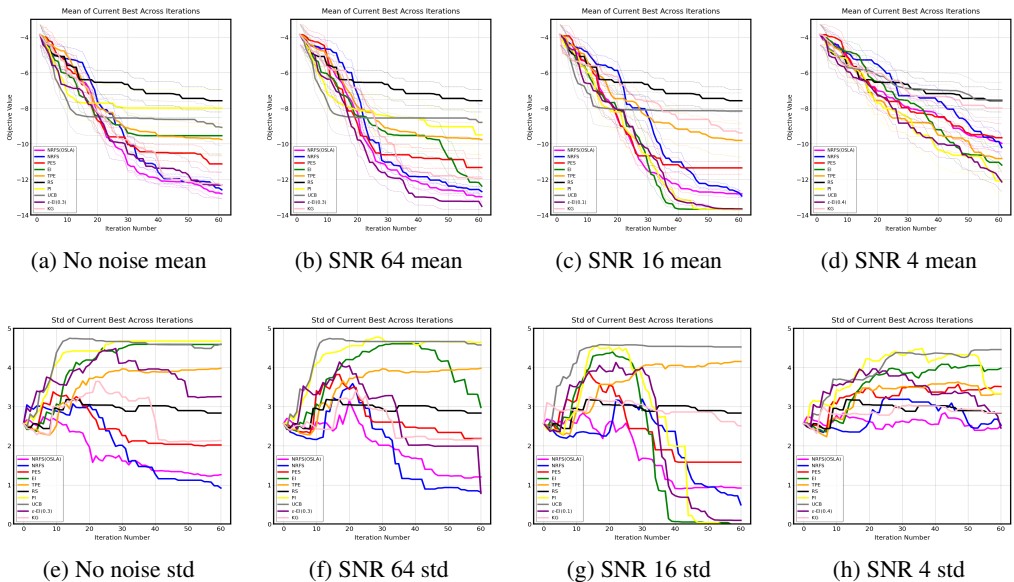

| (a) No noise mean | (b) SNR 64 mean | (c) SNR 16 mean | (d) SNR 4 mean |

| (e) No noise std | (f) SNR 64 std | (g) SNR 16 std | (h) SNR 4 std |

Figure 14: Performance comparison on modified Shekel function with different SNR levels.

robustness arises from TPE's use of a percentile-based threshold over top-performing observations, rather than relying solely on the previously observed best. In other words, with a given noise level, the deviation in the lower bound of the top-performing set is smaller than that of the previous best values. For instance, if the mean of observations is used as the threshold, the deviation between the noisy threshold and the true threshold becomes zero. However, this stability also reduces its ability to escape local optimal regions, thereby lowering the chance of identifying the global optimum.

The other methods are affected by noise in different ways. An interesting observation is that, after introducing noise, methods such as EI and PI can escape from regions of local optima. This is because the surrogate model becomes smoother with noise compared to the one learned in the corresponding noise-free case. This results in wider modes, making it easier for these methods to enter or escape such regions. However, the limitation is clear: only a small amount of noise helps prevent from oversampling. When the noise level is too high, it still leads to degraded performance. Since we are dealing with a black-box objective function, it is not possible to determine whether the added noise is within a safe or effective range. As a result, introducing noise cannot be considered a stable or reliable strategy to avoid oversampling.

## A.6 COMPUTATIONAL COST

For all objective functions, the experimental runs were distributed across 8 Intel(R) Xeon(R) Gold 6248R CPUs.

For the non-one-step-look-ahead methods, we optimize the acquisition function by evaluating it over all candidates and selecting the one with the maximum value.

For KG and one-step-look-ahead NRFS, if we denote the number of fantasy samples in the one-step-look-ahead step as $M$, then the computational complexity becomes $d^2M$, which incurs a high cost when computed over the entire design space. To mitigate this, we first use Latin Hypercube Sampling (LHS) (Shields & Zhang, 2016) to generate evenly distributed acquisition evaluation points (100) in the design space. The acquisition function is then computed on these points to identify the maximizer. Finally, we select the nearest neighbor of this maximizer from all the grid candidates and use the selected candidate as the next evaluation point. This strategy reduces computational resource consumption to a level comparable with other methods such as PES.

Table 1 reports the per-iteration computation time for each BO method with the corresponding acquisition function. The computational cost of NRFS is comparable to that of EI and remains stable across iterations. The additional overhead of the one-step-look-ahead NRFS arises primarily from

Table 1: Run-time (seconds) per iteration on different objective functions with different BO methods

| Acquisition function | GM | Modified Rosenbrock | Forrester2D | Shekel | Branin | BraninRcos2 |
|---|---|---|---|---|---|---|
| RS | 0.0003 | 0.0004 | 0.0003 | 0.0003 | 0.0003 | 0.0003 |
| TPE | 0.004 | 0.021 | 0.018 | 0.021 | 0.019 | 0.018 |
| $\epsilon$-EI | 1.073 | 0.721 | 0.703 | 0.654 | 0.699 | 0.711 |
| EI | 1.150 | 1.017 | 1.132 | 0.985 | 1.007 | 1.185 |
| PI | 0.372 | 0.942 | 0.739 | 0.724 | 0.846 | 0.730 |
| UCB | 0.256 | 0.776 | 0.592 | 0.603 | 0.702 | 0.659 |
| **NRFS** | 0.221 | 1.792 | 1.437 | 1.398 | 0.816 | 1.131 |
| PES | 8.014 | 11.245 | 10.458 | 9.671 | 8.931 | 10.120 |
| KG | 9.356 | 10.167 | 8.134 | 9.871 | 10.776 | 8.804 |
| **NRFS (OSLA)** | 5.432 | 7.114 | 5.437 | 5.557 | 5.981 | 5.102 |

repeated model updates. In our setup, we use $M = 5$ fantasy samples, resulting in the runtime approximately five times that of NRFS.

Among all methods, random sampling (RS) is the fastest, as expected. TPE exhibits relatively lower computation time compared to EI, primarily due to the efficiency of kernel density estimation relative to Gaussian process updates. PES incurs significantly higher computational cost; this is consistent with what has been reported in previous work, as PES typically relies on expectation propagation (EP) (Hennig & Schuler, 2012) to derive an analytical approximation of the utility function. For KG, the computational time increases over iterations: it starts at approximately 1 second per iteration and grows to around 15 seconds by iteration 60.

## A.7 USE OF LARGE LANGUAGE MODELS

Large Language Models are only used to check vocabulary and grammar for polishing purpose.

