# OpenReview forum: "Non-Replacement Function Space Sampling for Bayesian Optimization"
_ICLR.cc/2026/Conference — Submitted to ICLR 2026_

### Official Review · Reviewer_vSen · 2025-10-31

**Soundness:** 3
**Presentation:** 2
**Contribution:** 2
**Rating:** 2
**Confidence:** 3

**Summary:**

This paper considers the problem of black-box optimization, in which one is trying to optimize a function while having access only to noisy function calls: no gradient, convexity, or other information is available. In this setting, Bayesian Optimization (BO) is the state-of-the-art approach. BO learns a statistical surrogate for the objective and plugs it into an acquisition function (AF), whose maximization yields the next design to evaluate, in an exploration-exploitation balancing way. While other AFs often achieve this balance _via_ tedious hyperparameter tuning, the authors propose Non-Replacement Function Space Sampling (NRFS), which instead achieves this tradeoff by prioritizing sampling functions that have not been involved in previous design proposals. After deriving the practical computation steps for their AF, the authors evaluate against several other baselines and demonstrate its superiority on several test problems.

**Strengths:**

- The approach is theoretically motivated, and Figure 1 clearly highlights a failure mode of other AFs that the proposed method solves, on the example shown.
- The idea of directly involving $P(x^\star_{f^\star} \notin X_n)$ in the acquisition function is interesting and novel.
- Among the range of experiments performed, NRFS obtains the best performance.

**Weaknesses:**

- The original contribution of this paper may feel thin. To the best of my understanding, the proposed acquisition function, NRFS, boils down to that derived in [1], with an additional truncation? These feel quite close, even if the initial motivations can be different. It is completely fine for the proposed approach to be simple and close to other works, specifically when empirical performances are excellent. But here, the experiment section is too small to conclude so. In the absence of a strong experiment section, for this kind of work, I believe the contribution should also involve a theoretical analysis (e.g., regret bounds), which is also not present.

- More precisely, regarding the experiment section, I think more of them could have been conducted. State-of-the-art BO papers now compare proposed AFs to a larger range of settings. I will suggest some:
a. Batch acquisition setting
b. Denser high-dimensional experiment section: more examples, larger range of dimensionalities tested. Lots of synthetic test cases can be extended to any dimensionality (e.g., Rosenbrock, Ackley), such that one can consider for instance Ackley 10D, 20D, 50D... Even more simply, considering GP draws from a high-dimensional GP prior is fine.
There are also plenty of real-world high-dimensional cases (e.g., LassoDNA, Mopta08, Rover). Also, I do not think using SAASBO is always necessary, specifically for "very" high-dimensional settings where computations will become prohibitively expensive.
c. I would add Max-value Entropy Search (MES, [2]) to the other AFs being compared, as a more sensible representative of information-theoretic AFs, given that it often outperforms PES.

[1] Entropy Search for Information-Efficient Global Optimization, JMLR, 2012.
[2] Max-value Entropy Search for Efficient Bayesian Optimization, ICML 2017

**Questions:**

- Please define what you meant by _modified_ Rosenbrock function.
- What are the characteristics of the GP surrogate employed here? Which kernel is being used? How are its lengthscale and variance optimized? Are you using a lengthscale initialization that scales with the problem dimensionality as suggested by [3]? This is now the default in BoTorch, the library you mentioned for implementing the other acquisition functions, so I believe this would be the case; however, all this should be mentioned explicitly.
- Can you write the exact equation that is being used in your code once you sampled $M$ draws from the TGP? What is the grid these draws are evaluated on?

Additional comments:
- I believe the standard deviations presented in the regret plots are computed over 20 different seeds (Appendix A.3?): this should be stated in the main text directly, as well as whether you are plotting $\pm$1 or 2 stds.
- In Figure 2, using a time-varying color coding for the evaluations (or an increasing markersize) would give us a sense of how design space exploration is performed.

[3] Vanilla Bayesian Optimization Performs Great in High Dimensions, ICML 2024

I am willing to increase my score, provided that the weaknesses and questions stated above are addressed.

---

> ### Author Response · Authors · 2025-12-04
>
> ## Detailed Responses to Reviewer vSen
>
> We truly appreciate your constructive critiques and suggestions. In the following, we provide our point-by-point responses to address the raised concerns.
>
> ### **Weaknesses:**
>
> **In the absence of a strong experiment section, for this kind of work, I believe the contribution should also involve a theoretical analysis (e.g., regret bounds)**
>
> We are currently developing a new theoretical framework to establish the convergence of our acquisition function, but this work is unlikely to be completed within the rebuttal period. Therefore, we rely on additional experiments to strengthen the credibility of our method.
>
> Regarding regret analysis, our view is that existing proofs, which require all future acquisitions to converge to the global optimum, can lead to oversampling and inefficiencies. In our view, demanding that $x\_{n} \to x^\*$ is not an appropriate goal in black-box Bayesian optimization. Unlike transparent (“clear-box”) optimization, black-box settings must consider uncertainty; forcing $x\_{n} \to x^\*$ drives the acquisition uncertainty to zero prematurely. Classical results such as those for EI adopt this convergence of $x\_{n}$ perspective, which leads to the well-known trapping or over-sampling issues because uncertainty in future acquisitions collapses. To fundamentally avoid this oversampling problem, we redefine the goal: as long as at least one point in the previous observations $X_n$ converges to $x^\*$, we consider the optimizer to be found. We are currently working on new theratical framework to prove this new goal.
>
> More broadly, we believe that researchers in black-box BO should clearly state the specific goal their methods are designed to optimize. If the goal is finding the global optimizer, our formulation should be more appropriate. If the goal is maximizing performance at each individual iteration, then EI may be preferrable. These two goals are different, and prior regret analysis has not always treated them distinctly.
>
> **I will suggest some: a. Batch acquisition setting b. Denser high-dimensional experiment section: more examples, larger range of dimensionalities tested. Lots of synthetic test cases can be extended to any dimensionality c. I would add Max-value Entropy Search (MES, [2]) to the other AFs being compared, as a more sensible representative of information-theoretic AFs, given that it often outperforms PES.**
>
> We have added: (a) batch acquisition experiments with batch sizes of 2, 5, and 10, comparing their performance to sequential optimization in the main text; (b) additional benchmark tasks, including high-dimensional Forrester functions with 5, 10, 20, and 50 dimensions in the main text, and real-world high-dimensional cases such as Lasso-DNA, Rover, and Mopta08 in the supplementary material; and (c) four extra baseline methods: MES, JES, Thompson Sampling (TS), and Probability of Optimality (PO).
>
> We chose the Forrester function over Ackley because Ackley resembles a single large mode with small additional modes that act like noise, whereas Forrester provides multiple valleys, representing more complex landscapes. The results for Lasso-DNA, Rover, and Mopta08 are included in the supplementary material, as our main goal is to demonstrate that our acquisition methods avoid the limitations of previous strategies, rather than to benchmark performance in high-dimensional settings. In very high-dimensional tasks, such as Mopta08 (with a design space of size $10\^{180}$), exploration is extremely inefficient because uncertainty remains high everywhere regardless of the number of acquisitions. Consequently, there is effectively no exploration–exploitation trade-off, and focusing resources on exploitation can be more efficient, as illustrated in Supplementary Figure 10.

---

> > ### Author Response · Authors · 2025-12-04
> >
> > ### **Questions:**
> >
> > **Please define what you meant by modified Rosenbrock function.**
> >
> > The modified Rosenbrock function is defined as
> > $f(x_1, x_2) = 100 (x_2 - x_1^2)^2 + (1 - x_1)^2 - 400 \exp\Biggl(-\frac{(x_1 + 1)^2 + (x_2 + 1)^2}{0.1}\Biggr)$. We have added the reference for the source of this function.
> >
> > **What are the characteristics of the GP surrogate employed here? Which kernel is being used? How are its lengthscale and variance optimized?**
> >
> > Yes, we use the BOTorch setup for the GP surroagte initialziation and fitting. The detail can be found in the supplementry.
> >
> > **Can you write the exact equation that is being used in your code once you sampled $M$ draws from the TGP?**
> > The simple answer is $x^* = \mathbb{E}\_{f \sim \mathcal{F}} \Big[ \arg\max\_{x \in \mathcal{X}} f(x) \Big]$.
> >
> > The overall idea is to identify the optimziers of these $M$ sampled functions $\\{f\_j(x)\\}\_{j=1}^M$ and identify the most possible optimizer location. For example if we have a discrete case as our real case material inverse design. The grid is just all the possible compositions, $\\{z\_l\\}\_{l=1}^N \subset \mathcal{X}$. We then treat $z^*\_{f_j} = \arg\min\_{l\in\\{1,\dots,N\\}} f\_j(z\_l)$ as the true optimizer of function $f\_j$. Next we compute how frequently each $z\_l$ is selected as the optimizer across $M$ function samples, and select the point with the highest frequency as the next acquisition.
> >
> >
> > **I believe the standard deviations presented in the regret plots are computed over 20 different seeds (Appendix A.3?): this should be stated in the main text directly.**
> >
> > Thanks for pointing this out, we have updated the setup description in our main text.
> >
> > **In Figure 2, using a time-varying color coding for the evaluations (or an increasing markersize) would give us a sense of how design space exploration is performed.**
> >
> > We have updated Figure 2. It shows that while most of the later acquisitions concentrate on the global optimum region, NRFS also occasionally samples high-uncertainty areas, allowing it to avoid missing potential optima in unexplored regions.
> >
> > **I am willing to increase my score, provided that the weaknesses and questions stated above are addressed.**
> >
> > We thank the reviewer for providing the possibility for score increase. We have completed all the experiments mentioned in this review, including: Batch acquisition analysis with batch sizes 2, 5, and 10 on 4 synthetic objectives. Additional baselines: MES, JES, Thompson Sampling (TS), and Probability of Optimality (PO) on all 6 main benchmark objectives in the orignal main text. Extra high-dimensional benchmarks on the Forrester function with 5, 10, 20, and 50 dimensions for all 14 baselines and three real-world cases: Lasso-DNA, Rover, and Mopta08.
> >
> > While the results on the high dimensional real-world cases do not show dominant performance, this is expected: our method is designed to achieve an optimal balance between exploration and exploitation, aiming for the global optimum rather than rapid exploitation in high-dimensional spaces.
> >
> > We would greatly appreciate any acknowledgment of the effort invested in conducting these comprehensive experiments.

---

### Official Review · Reviewer_RS1Z · 2025-10-31

**Soundness:** 2
**Presentation:** 2
**Contribution:** 3
**Rating:** 4
**Confidence:** 4

**Summary:**

The authors propose an acquisition function which is a modification of Predictive Entropy Search (PES). For PES, the probability of an optimal location for an input must be estimated via drawing samples from a Gaussian process surrogate. The authors propose to assess the utility of a point by how many possible surrogate functions would model it as the optimum. In addition, they also provide a one-step-lookahead formulation. They evaluate their method on low-dimensional synthetic functions and two material design tasks, showing that it exhibits strong anytime and final performance.

**Strengths:**

As the authors state, their approach presents a novel perspective on acquisition functions. To this end, they provide theoretically well-motivated derivations of their approach and combine it with practical implementation ideas, such as truncating the Gaussian process (GP) to exclude samples that do not contain an optimum better than the current known one.

The line of thought is well presented by grouping the GP samples for which x is a minimizer, and by connecting the Probability of Optimality to the function space coverage ratio.

An intuition of the effectiveness and the exploration-exploitation trade-off (EETO) is well demonstrated by showing convergence performance on synthetic test functions and two real-world problems.

**Weaknesses:**

**Novelty**
L074 The claim of a “new probabilistic framework” is, in my opinion, overstated when the method is essentially a different view on Entropy Search but with a Truncated GP. Please adapt the claim accordingly. Also clarify the fact that Probability of Optimality (PO) is the formula from Hennig & Schuler \[2012\].
Furthermore, it is yet another acquisition function of BO. I disagree with the authors that their acquisition function does not encode the exploration-exploitation trade-off (EETO). The inherent design of black-box optimization with an unknown required EETO necessarily leads to an EETO. I would even argue that since NRFS only implicitly encodes the EETO, it is less flexible than other acquisition functions.

**Clarity**
Overall, the clarity is okay, but there are several smaller issues:

- L050: Depending on the community, the wording “optimizer” refers to the optimization method. It would be beneficial to introduce the term.
- Eq 2: Define that you are minimizing in your setup earlier.
- L101: The wording is very unfortunate, please update. AFs like EI and PI do not “tune” the EETO explicitly during the optimization process, but their EETO is encoded via the AF definition. Your method also has an EETO, which is also implicitly encoded and not set via a hyperparameter.
- L126: The kernel must be positive-semidefinite, thus it definitely does not map into R.
- L133: Unfortunate wording. The term utility comes from Bayesian decision theory. For some AFs, like EI and ES, the AF is actually the expected *increase* in utility (see Garnett Book 2023 for a compact description).
- L211&214 : Be consistent in your notation. Where does Y and I come from?
- L366: Introduce the name OSLA.
- Figures: The fontsizes are way too small. The rule is, if you need to zoom in to read it, it is too small. Please increase the fontsize and use a colorblind-friendly color palette.

**References /  Related Work**
Overall, I was missing many references to statements made in the related work sections:

- If you cite VES (newer ES methods), then also cite JES  \[Hvarfner et al., 2022\]. In addition, in the third paragraph of related work (starting L171) all references are missing. Please fix that.
- Also, where does ε-EI come from?
- For the online tuning strategies cite also \[Kushner 1964; Mockus: Bayesian Approach to Global Optimization 1989; Srinivas et al.: GP UCB 2010; Hoffman et al 2011 Portfolio Allocation; Benjamins et al.: Self-Adjusting Weighted Expected Improvement for Bayesian Optimization 2023\] and also the sources where you got the numbers 50-100 from.
- Additionally, I disagree with your last paragraph of related work. First of all, different tasks require different settings of BO \[Lindauer et al., 2019\]. Second, what is the definition of universal? That all kinds of search spaces can be searched in, like mixed-space, fully discrete, with conditions and constraints? What does “works” mean? Performing better than random? Performing better than all existing methods on average? On which tasks/domains? One can argue that the estimation of PO is also a heuristic. As your method is modified ES, which has been classified as guided by subjective reward, the paragraph does not positively impact the presentation of the method.

**Experimental Setup**
In general, the empirical evaluation is fairly limited in comparison to the strong claims of the authors. These are mainly four cherry-picked artificial functions and two real-world benchmarks. State-of-the-art evaluations show performance on many more benchmark tasks (e.g., BBOB for many more artificial functions or YAHPOGym or CARPS for extensive black-box HPO problems).
Furthermore, I’m irritated that the authors first show sampling behavior of NRFS on Branin, BraninRcos2, Himmelblau and HolderTable, but then use a different set of functions for the comparisons to the approaches. Additionally, the argument that these functions “require a balance of exploration and exploitation to effectively locate the global optimum” suggests to me that they deliberately sought functions that fit their method.
Also, important details for reproducibility are missing, e.g., how many random seeds were used, what exactly is actually plotted in Figure 3 and the dimensionality of SFE and HC.

Hidden in the appendix, another problem is obvious. While most established methods are fairly efficient to compute, NRFS with lookahead is 5 to 7 times more expensive to compute (Table 1 in the appendix).

**Questions:**

* What do you think, how would the model recover from model misspecification aka an unfortunate kernel choice?
* L064: How did you position the acquisition functions (AFs) on the EETO scale?
* What are the thin lines in the plots? The std or 95%-CI?
* L309: What is this hyperparameter gamma, and where does it come from?
* L423: Where does this schedule come from? Is that the standard one from GP-UCB?
* Have the experiments been repeated? What is the number of seeds?
* What about functions that require more exploration and vice versa?
* What is the dimensionality of all objective functions?
* How did you determine the optimization resources in terms of objective function evaluations?
* Please compare your method more thoroughly to Thompson sampling (in text). What is the impact of the number of samples? In addition, how important is the estimation technique (L332)?

---

> ### Author Response · Authors · 2025-12-04
>
> ### **Summary:**
>
> **The authors propose an acquisition function which is a modification of Predictive Entropy Search (PES).**
>
> Our method is not based on Predictive Entropy Search (PES). Sampling directly from a distribution is fundamentally different from reducing its entropy. We have already explained in the main text, both conceptually and through experiments, that PES tends to suffer from limited exploitation, and that our approach avoids this issue because it relies on direct sampling. To avoid redundancy, we will not repeat those arguments here.
>
> We would greatly appreciate it if the reviewer could show us where in our manuscript we indicated that our method is a modification of PES. This would help us clarify and revise the paper accordingly. Without a shared understanding of the basic mechanism of our method, it is difficult for us to constructively address this concern in the rebuttal.

---

### Official Review · Reviewer_9jUU · 2025-11-01

**Soundness:** 3
**Presentation:** 2
**Contribution:** 2
**Rating:** 4
**Confidence:** 3

**Summary:**

The paper proposes the Non-Replacement Function Space (NRFS) acquisition function for Bayesian optimization. The main idea is to directly estimate the probability of an input x to be a global optimum x*, given a dataset D_t observed up to iteration t. This probability P(x=x*|D_t) is written in Eq. 12, which can be computed by rejection sampling from the truncated Gaussian process posterior. Then, the authors further derive the one-step look-ahead (OSLA) variant of NRFS, to enhance the acquisition efficiency. The proposed acquisition function is evaluated against other common AF in BO on a set of synthetic and real-world benchmark problems.

**Strengths:**

-	The proposed acquisition function focuses on the main purpose of optimization problems – finding the local optimum, without having to use additional reward functions.
-	The proposed approach to optimize the NRFS acquisition function is sound.

**Weaknesses:**

1.	Even though the idea is sound, I think the implementation (the optimization of Eq. 12, the core of NRFS) is neither efficient nor guaranteed to work reliably. First, sampling M=1000 realizations from the truncated GP is not always efficient. This is because when the optimization identifies a promising solution Y*_n, drawing from the truncated distribution for samples better than the threshold Y*_n becomes increasingly difficult and computationally expensive. Second, the procedure for finding the optimizer of each realization is non-trivial but insufficiently described. It is unclear whether the authors use gradient-based optimization over a continuous realization function or rely on a discretized grid search. Nevertheless, both approaches pose challenges due to the multi-modality of the sampled realizations: gradient-based methods require multiple restarts, whereas grid-based searches require high density, increasing computational cost.
2.	Additionally, optimizing Eq. 16 (the OSLA variant) is also computationally expensive. Since the authors propose to generate M_s fantasy samples and re-compute NRFS M_s times, the total runtime scales linearly with M_s (as confirmed by the run time analysis). In the experiments, although the authors use 5 fantasy samples, which seems feasible, this number may not sufficiently approximate the expectation term in Eq. 16. If more fantasy samples are needed for accuracy, the computational burden would increase substantially.
3.	Apart from the PES baseline (2014), the paper lacks newer state-of-the-art baselines from the Entropy Search techniques, such as MES (2017) [1] and JES (2022) [2]. Additionally, Thompson Sampling [3] needs to be included.
4.	In my opinion, Eq. 16 should not be called the “one-step look-ahead variant” of Eq. 12. Both equations are one-step look-ahead: they both consider the utility of the next decision. The difference is that Eq. 12 does not consider the outcome y while Eq. 16 takes y into account, hence resulting in a more complex variant.

[1] Wang, Zi, and Stefanie Jegelka. "Max-value entropy search for efficient Bayesian optimization." International conference on machine learning. PMLR, 2017.

[2] Hvarfner, Carl, Frank Hutter, and Luigi Nardi. "Joint entropy search for maximally-informed Bayesian optimization." Advances in Neural Information Processing Systems 35 (2022): 11494-11506.

[3] Thompson, William R. "On the likelihood that one unknown probability exceeds another in view of the evidence of two samples." Biometrika 25.3/4 (1933): 285-294.

**Questions:**

1.	It is surprising that Random Search is so much better than some common acquisition strategies. In Fig. 3, RS outperforms PI on 5/6 problems, TPE on 4/6 problems, EI on 3/6 problems, KG on 2/6 problems and PES in 1/6 problem. It has been well-known that these acquisition functions are at least better than RS, especially on these low-dimensional problems, so I expect RS to be the worst-case scenario on all benchmark problems. This poses a significant concern about the credibility of the reported experimental results. Can the authors comment on this behavior?
2.	In lines 97-99, the authors mention to “remove all functions … from future consideration…”. How exactly can a function be removed from being sampled by the TGP?
3.	In Appendix A.6, what are the “non-one-step-look-ahead methods”? Is it the “standard” NRFS (Eq .12) only, or does it include other baselines? Additionally, how did the authors handle EI and PI, as they are one-step look-ahead acquisition functions, but were not mentioned in the section?
4.	How do the NRFS acquisition function and its OSLA variant perform in larger budget settings? The current experiments show up to 60 iterations, so given the Weaknesses 1 and 2, I wonder if there are any computational cost issues when running with more iterations, e.g., up to 200 or 500?

---

> ### Author Response · Authors · 2025-12-04
>
> ## Detailed Responses to Reviewer 9jUU
>
> We truly appreciate your constructive critiques and suggestions. In the following, we provide our point-by-point responses to address the raised concerns.
>
> ### **Weaknesses:**
>
> **Even though the idea is sound, I think the implementation (the optimization of Eq. 12, the core of NRFS) is neither efficient nor guaranteed to work reliably.**
>
> Thank you for pointing out this technical issue. In fact, it was one of the first concerns we considered when developing our strategy. Fortunately, prior work has shown that truncated Gaussian distributions can be sampled directly using efficient, non-rejection algorithms [1, 2]. Thus, choosing $M=1000$ ensures that all 1000 samples are genuine draws from the truncated Gaussian distribution.
>
> We also examined whether this approach remains applicable in future MOBO settings. Even when the truncation region is defined by a complex Pareto front, direct sampling is still feasible by applying a box-decomposition of the dominated region. In our experiments, we use the well-established implementation provided in the scipy package for truncated Gaussian sampling [3].
>
> **It is unclear whether the authors use gradient-based optimization over a continuous realization function or rely on a discretized grid search.**
>
> We designed different strategies depending on the problem type. As noted in Section 4.1, when the search space is continuous, we apply a Parzen estimator to obtain a smooth density function, and then use a gradient-based method to locate its optimizer. For the real-case benchmarks in Section 4.3, where the compositions are discretized, we instead sample discretized realization functions and estimate the frequency with which each composition becomes the optimizer. Because optimizing a known, explicit function is substantially easier than black-box optimization, we did not include these acquisition function evaluation details in the main text. We will add these experimental setup descriptions in the revised version.
>
> **The computational burden would increase substantially for OSLA.**
>
> As noted earlier, the sampling issue is avoided because we do not rely on rejection sampling. Although the computational complexity is relatively high for all OSLA-based methods, Table 1 in Appendix A.6 shows that the runtime of NRFS remains lower than that of PES and KG. Moreover, we are confident that this cost can be further reduced through analytical approximations. We are actively exploring approaches to derive a closed-form or near–closed-form approximation for NRFS. For example, we can predict where the optimum might be for each observation location and evaluate optimality based on the expected optima across all locations. This approach may yield an analytical version of NRFS with a structure similar to EI.
>
> More importantly, in real-world applications, the dominant cost arises from the experiments or simulations themselves rather than from the computational overhead of NRFS or other MOBO methods. Real evaluations typically take far longer than a few seconds, for example, as discussed in the main text for materials discovery, when the task is not a rediscovery case, fabricating and testing a new composition can take weeks or even months. Therefore, the computational cost considered here is not a significant concern.
>
> **The paper lacks newer state-of-the-art baselines from the Entropy Search techniques, such as MES, JES and TS.**
>
> We included MES, JES, PO and TS as additional baselines to benchmark. In our previous main results, we primarily use PES because it targets the coverage of the design space rather than the objective space, as our NRFS. In contrast, MES and JES focus more on the objective space, making them less aligned with the goal of NRFS. For this reason, we did not select MES or JES as the primary representatives of the exploration-oriented acquisition class. For Thompson Sampling (TS), we treat it as a one-shot ($M=1$) Probability of Optimality (PO). Therefore, we believe it exhibits the same oversampling behavior that PO suffers from, as illustrated in Fig. 1.

---

> > ### Author Response · Authors · 2025-12-04
> >
> > **In my opinion, Eq. 16 should not be called the “one-step look-ahead variant” of Eq. 12.**
> >
> > In the main text, we classify methods that explicitly incorporate a hypothetical future observation into the surrogate model as one-step-look-ahead approaches. In contrast, methods such as EI and PI only evaluate the expected value of potential outcomes. They do not condition on a GP refitted with an augmented dataset that includes each possible future observation. Therefore, EI and PI do not perform the same type of look-ahead reasoning as the one-step-look-ahead methods we discuss.
> >
> >
> > ### **Questions:**
> >
> > **It is surprising that Random Search is so much better than some common acquisition strategies.**
> >
> > As you mentioned, for single modal or smooth objective functions, EI and similar methods can easily outperform random sampling (RS). However, since our goal is to develop an implicit acquisition method that best balances exploration and exploitation, benchmarking on these easy tasks does not reveal the key issues we aim to address, such as oversampling and low exploitation efficiency. To evaluate these aspects, it is necessary to focus on more complex, multi-modal problems as we discussed in Section 4.3. On these difficult cases, methods like TPE, EI, KG, and PI all suffer from similar oversampling behavior, which leads to performance that can be even worse than RS [4]. PES performs better than these exploitation biased methods on such complex problems, but its lower exploitation efficiency can still cause it to underperform RS in some situations [5].
> >
> > **How exactly can a function be removed from being sampled by the TGP?**
> >
> > By “remove,” we mean that any sampled function whose optimal value is already worse than the current best is discarded and will never be sampled again.
> >
> >
> > **In Appendix A.6, what are the “non-one-step-look-ahead methods”.**
> >
> > As mentioned earlier, we define methods that add a hypothetical future observation to the current dataset and then refit a surrogate GP as non–one-step-look-ahead methods. Under this definition, PES, KG, and NRFS (OSLA) fall into the category of non–one-step-look-ahead approaches. The computational cost of these three methods are higher compared to other methods.
> >
> >
> > [1] https://arxiv.org/abs/1201.6140
> >
> > [2] https://link.springer.com/article/10.1080/15598608.2014.996690
> >
> > [3] https://docs.scipy.org/doc/scipy/reference/generated/scipy.stats.truncnorm.html
> >
> > [4] https://arxiv.org/abs/1911.07285
> >
> > [5] https://arxiv.org/abs/2306.00344

---

### Meta-Review · Area_Chair_Vht5 · 2026-01-11

**Summary:**

This paper considers the problem of solving black-box optimization problems within the framework of Bayesian optimization (BO). The paper develops a new acquisition strategy referred as Non-Replacement Function Space (NRFS) which is based on the estimatation of the probability of a candidate input to be a global optimum. A one step lookahead variant of NRFS is also derived. Experiments are performed on synthetic benchmarks to demonstrate the efficacy of the proposed approach.

The reviewers' acknowledge the contribution of the paper in terms of a new perspective, but also raised some critical concerns:
1. Implementation and efficiency of acquisition function and its optimization.
2. Lack of recent baselines such as MES and JES.
3. Clarification on experimental results (e.g., random search performing much better) and benchmark functions.
4. Novelty claims and related work

The rebuttal from authors' addressed some of the concerns satisfactorily (#1 and #2) and less satisfactorily others (#3 and #4). The AC took a look at the revised paper and feel that the experimental evaluation is not convincing:
(a) the benchmarks were limited (as pointed by couple of reviewers). The surprising performance of random search pointed out by one reviewer and the authors' response suggest a broader evaluation is needed.
(b) some of the newer experiments didn't support the efficacy of the method -- authors' write "While the results on the high dimensional real-world cases do not show dominant performance, this is expected: our method is designed to achieve an optimal balance between exploration and exploitation, aiming for the global optimum rather than rapid exploitation in high-dimensional spaces."
(c) authors' didn't respond to questions from Reviewer RS1Z (many of them are valid).

For the above reasons, I recommend rejecting the paper and strongly encourage the authors to improve the paper for future re-submission.

**Reviewer Concerns:**

The reviewers' acknowledge the contribution of the paper in terms of a new perspective, but also raised some critical concerns:
1. Implementation and efficiency of acquisition function and its optimization.
2. Lack of recent baselines such as MES and JES.
3. Clarification on experimental results (e.g., random search performing much better) and benchmark functions.
4. Novelty claims and related work

The rebuttal from authors' addressed some of the concerns satisfactorily (#1 and #2) and less satisfactorily others (#3 and #4). The AC took a look at the revised paper and feel that the experimental evaluation is not convincing:
(a) the benchmarks were limited (as pointed by couple of reviewers). The surprising performance of random search pointed out by one reviewer and the authors' response suggest a broader evaluation is needed.
(b) some of the newer experiments didn't support the efficacy of the method -- authors' write "While the results on the high dimensional real-world cases do not show dominant performance, this is expected: our method is designed to achieve an optimal balance between exploration and exploitation, aiming for the global optimum rather than rapid exploitation in high-dimensional spaces."
(c) authors' didn't respond to questions from Reviewer RS1Z (many of them are valid).

**Reviewer Scores:**

Reviewer 9jUU: 4 => 5
Reviewer RS1Z: 4
Reviewer vSen: 2 => 4

---

### Decision · Program_Chairs · 2026-01-26

Reject